# Interactions between Membrane Resistance, GABA-A Receptor Properties, Bicarbonate Dynamics and Cl^−^-Transport Shape Activity-Dependent Changes of Intracellular Cl^−^ Concentration

**DOI:** 10.3390/ijms20061416

**Published:** 2019-03-20

**Authors:** Aniello Lombardi, Peter Jedlicka, Heiko J. Luhmann, Werner Kilb

**Affiliations:** 1Institute of Physiology, University Medical Center Mainz, Johannes Gutenberg University, Duesbergweg 6, 55128 Mainz, Germany; alombard@uni-mainz.de (A.L.); luhmann@uni-mainz.de (H.J.L.); 2ICAR3R - Interdisciplinary Centre for 3Rs in Animal Research, Faculty of Medicine, Justus-Liebig-University, Rudolf-Buchheim-Str. 6, 35392 Giessen, Germany; Peter.Jedlicka@informatik.med.uni-giessen.de; 3Institute of Clinical Neuroanatomy, Neuroscience Center, Goethe University, 60590 Frankfurt am Main, Germany; 4Frankfurt Institute for Advanced Studies, 60438 Frankfurt am Main, Germany

**Keywords:** development, hippocampus, CA3, Cl^−^-homeostasis, giant depolarizing potentials, ionic plasticity, computational neuroscience, Na^+^-K^+^-Cl^−^-Cotransporter, Isoform 1 (NKCC1), mouse

## Abstract

The effects of ionotropic γ-aminobutyric acid receptor (GABA-A, GABA_A_) activation depends critically on the Cl^−^-gradient across neuronal membranes. Previous studies demonstrated that the intracellular Cl^−^-concentration ([Cl^−^]_i_) is not stable but shows a considerable amount of activity-dependent plasticity. To characterize how membrane properties and different molecules that are directly or indirectly involved in GABAergic synaptic transmission affect GABA-induced [Cl^−^]_i_ changes, we performed compartmental modeling in the NEURON environment. These simulations demonstrate that GABA-induced [Cl^−^]_i_ changes decrease at higher membrane resistance, revealing a sigmoidal dependency between both parameters. Increase in GABAergic conductivity enhances [Cl^−^]_i_ with a logarithmic dependency, while increasing the decay time of GABA_A_ receptors leads to a nearly linear enhancement of the [Cl^−^]_i_ changes. Implementing physiological levels of HCO_3_^−^-conductivity to GABA_A_ receptors enhances the [Cl^−^]_i_ changes over a wide range of [Cl^−^]_i_, but this effect depends on the stability of the HCO_3_^−^ gradient and the intracellular pH. Finally, these simulations show that pure diffusional Cl^−^-elimination from dendrites is slow and that a high activity of Cl^−^-transport is required to improve the spatiotemporal restriction of GABA-induced [Cl^−^]_i_ changes. In summary, these simulations revealed a complex interplay between several key factors that influence GABA-induced [Cl]_i_ changes. The results suggest that some of these factors, including high resting [Cl^−^]_i_, high input resistance, slow decay time of GABA_A_ receptors and dynamic HCO_3_^−^ gradient, are specifically adapted in early postnatal neurons to facilitate limited activity-dependent [Cl^−^]_i_ decreases.

## 1. Introduction

GABA (γ-aminobutyric acid) is the main inhibitory neurotransmitter in the mature brain and acts via ionotropic GABA_A_/GABA_C_ receptors and via metabotropic GABA_B_ receptors [1]. In the adult brain, GABA mediates its inhibitory effect by hyperpolarizing the membrane and by shunting excitatory inputs. GABA_A_ receptors are ligand-gated anion-channels with a high permeability for Cl^−^ ions and a considerable additional permeability for HCO_3_^−^ ions [1]. In the mature brain the activity of a K^+^-Cl^−^-Cotransporter (KCC, mainly in its isoform KCC2) establishes a low intracellular Cl^−^ concentration ([Cl^−^]_i_) [2,3], which accounts for a Cl^−^ influx and thus a membrane hyperpolarization upon activation of GABA_A_ receptors [1]. Due to this Cl^−^-flux, activation of GABA_A_ receptors can influence [Cl^−^]_i_ on a time scale of seconds to minutes [4,5,6,7,8,9], a process termed “ionic plasticity” [3,10,11]. The magnitude of activity-dependent [Cl^−^]_i_-transients depends on the Cl^−^ influx, dendritic volume and morphology, as well as on the capacity of Cl^−^ extrusion systems [12,13,14,15,16,17]. In addition, the membrane potential and the HCO_3_^−^ permeability of GABA_A_ receptors (P_HCO_3__) contribute to the size of [Cl^−^]_i_ changes [6,18,19,20]. Therefore recent concepts of inhibition considered neuronal [Cl^−^]_i_ as a state- and compartment-dependent parameter of individual cells [14,20]. Detectable activity-dependent [Cl^−^]_i_ changes can occur in the adult nervous system under massive GABAergic stimulation [12,21]. However, already small alterations in [Cl^−^]_i_ or in the dynamics of the [Cl^−^]_i_ homeostasis critically influence information processing in neurons [20,22]. As the proper function in the adult nervous system relies on adequate inhibition [1,23,24], these activity-dependent [Cl^−^]_i_ changes play important roles in physiological and pathophysiological processes [10,11,17].

In the immature nervous system GABA typically induces depolarizing membrane responses [25,26,27,28,29,30]. These depolarizing GABAergic responses are caused by an elevated intracellular Cl^−^ concentration ([Cl^−^]_i_), which is maintained by a Cl^−^ accumulation via the isoform 1 of the Na^+^-dependent K^+^-Cl^−^-cotransporter (NKCC1) [3,29,31,32]. Recent studies suggest that at least in the postnatal neocortex, these depolarizing GABAergic responses mainly mediate inhibition [30,33], likely by increasing membrane shunting [1,34]. Several results indicate that depolarizing GABAergic neurotransmission is of specific relevance for immature spontaneous activity and for the maturation of the central nervous system [35,36,37]. Giant depolarizing potentials (GDPs) are a well-described network phenomenon in the immature hippocampus and the neocortex that represent spontaneous GABA-dependent activity [25,38,39]. In line with the high [Cl^−^]_i_ and the depolarizing responses, activation of GABA_A_ receptors causes a decline in [Cl^−^]_i_ of immature neurons [29,40,41,42]. This attenuation of the [Cl^−^] gradient reduces possible excitatory effects of GABA [29,42,43] and may serve to limit GABAergic excitation and/or to stabilize recurrent network events [10,13,40]. A recent study demonstrated that GDPs, which are associated with a high amount of GABAergic activity [25,44], induce long-lasting [Cl^−^]_i_ transients and influence the steady-state [Cl^−^]_i_ of CA3 pyramidal neurons in hippocampal slices from early postnatal mice neurons [45], making it a suitable model for ionic plasticity in the immature brain.

However, while the existence of ionic plasticity is well accepted and several factors influencing activity-dependent [Cl^−^]_i_ transients have been described, the role of biophysical membrane characteristics, molecular properties of GABA_A_-receptors or Cl^−^-transporters, and the stability of HCO_3_^−^ homeostasis on ionic plasticity has not yet been systematically investigated. Here we used a detailed biophysical compartmental model in the NEURON environment to demonstrate how cellular and molecular properties such as input resistance, pH, HCO_3_^−^-selectivity, kinetics of GABA_A_ receptors, the kinetics of NKCC1 mediated Cl^−^ transmembrane transport, and the activity of carbonic anhydrases influence activity-dependent [Cl^−^]_i_ transients. While most modeling is performed in isolated dendritic compartments, here we also replicate the well-described GDP-induced [Cl^−^]_i_ transients of immature hippocampal CA3 neurons [45].

## 2. Results

In order to study the question how various membrane parameters and the properties of different molecules involved in GABAergic transmission influence activity-dependent [Cl^−^]_i_ transients, we first computed the GABA-induced [Cl^−^]_i_ changes in an isolated dendrite, which allows a better mechanistic understanding of the underlying processes. Subsequently we also used a model of a reconstructed CA3 pyramidal neuron [45] to compare the results of our computational models with the GDP-dependent [Cl^−^]_i_ transients recorded in immature hippocampal CA3 neurons [45]. For the latter model, we implemented experimentally derived parameters of GABAergic synapses and GDP-activity provided by Lombardi et al. [45].

### 2.1. Influence of Membrane Conductance

First, we analyzed the influence of the membrane conductance on the [Cl^−^]_i_ changes induced by a single GABAergic input in an isolated dendrite. In this model, the experimentally determined conductance underlying single spontaneous GABAergic postsynaptic responses (g_GABA_) was implemented in an isolated dendrite equipped with passive conductances (g_pas_) varying between 10^−6^ S/cm^2^ and 0.1 S/cm^2^. These passive conductances correspond to input resistances (R_Input_) between ca. 670 MΩ and 0.67 kΩ, respectively, when they were implemented in a reconstructed CA3 pyramidal neuron. The results of this experiment demonstrated that upon stimulation of a single GABAergic input (g_GABA_ = 0.789 nS, τ = 37 ms, P_HCO_3__ = 0, [Cl^−^]_i_ = 30 mM) not only the depolarization, but also the GABA-induced [Cl^−^]_i_ transient depended on the passive conductances. A detailed analysis revealed a strong, sigmoidal dependency between R_Input_ and peak [Cl^−^]_i_ changes (Figure 1a) or depolarization (Figure 1b) upon a single GABA stimulus. This effect of g_pas_ on the GABA-induced [Cl^−^]_i_ transients was caused by the fact that at lower g_pas_ the GABAergic currents induced a substantial depolarization, which attenuated the electromotive force on Cl^−^ ions (DF_Cl_) during GABA stimulation (Figure 1c). At a low g_pas_ of 10^−7^ S/cm^2^ (corresponding to a R_input_ of ca. 4 GΩ in the reconstructed neuron) the GABAergic depolarization reached E_Cl_ (Figure 1c, solid lines). Therefore, DF_Cl_ approximated 0 and no persistent Cl^−^ fluxes occurred. In contrast, at a g_pas_ of 0.018 S/cm^2^ (corresponding to R_input_ of ca. 41 MΩ) E_m_ remained negative to E_Cl_, thus enabling permanent Cl^−^ fluxes (Figure 1c, dashed lines).

Next we simulated how g_pas_ influences [Cl^−^]_i_ in a reconstructed neuron (Figure 2a,b), which receives complex GABAergic inputs that typically occur during GDP activity [45] (Figure 2c,d). For these experiments we initially equipped the dendrite with 101 GABAergic synapses (g = 0.789 nS, τ = 37 ms; all values from Lombardi et al. [45]), set P_HCO_3__ to 0 and used an initial [Cl^−^]_i_ to 30 mM. Each of these 101 GABAergic synapses was randomly distributed within the dendrites of the reconstructed neuron. The time points for the stimulation of every synapse follows a normal distribution (µ = 600ms, σ = 900 ms). These values were derived from in-vitro experiments and resemble the distribution of GABAergic inputs during a GDP [45] (Figure 2c). In order to reduce the complexity of the analysis and to mimic the procedures of [Cl^−^]_i_ estimation used by Lombardi et al. [45] (which estimated [Cl^−^]_i_ changes from changes in E_Rev_ determined by focal GABA application within the dendritic compartment) we use for all further analyses the average [Cl^−^]_i_ of all dendrites. 

Using this model, we investigated how different g_pas_ between 10^−6^ S/cm^2^ and 0.1 S/cm^2^ affect the GDP-induced [Cl^−^]_i_ transients. This simulation demonstrated that also in a complex dendritic compartment g_pas_ critically influenced the amount of [Cl^−^]_i_ changes (Figure 2e). Also, under these conditions the GABAergic depolarization during a GDP approached E_Cl_ at low g_pas_ (Figure 2f), which minimized DF_Cl_ and the remaining Cl^−^ fluxes. One particular result of this computational study was that the GDP-induced [Cl^−^]_i_ transient amounts to less than 1 mM in a reconstructed CA3 pyramidal neuron equipped with the passive membrane conductance determined experimentally in these neurons (red symbols in Figure 2e–g), which is lower than the experimentally determined [Cl^−^]_i_ changes of 10.3 ± 3.3 mM (*n* = 4) in a real CA3 pyramidal neuron at comparable conditions [45]. To further specify the influence of g_pas_ on the GDP-induced [Cl^−^]_i_ transients, we simulated the peak dendritic [Cl^−^]_i_ change for different initial [Cl^−^]_i_ at three different g_pas_. For this purpose we used values of 0.049 mS/cm^2^ (corresponding to a R_Input_ of 901 MΩ, typical for immature hippocampal neurons [45]), 0.28 mS/cm^2^ (189 MΩ, adult neuron in whole-cell patch-clamp configuration [46]), and 1.8 mS/cm^2^ (41 MΩ, adult neuron with sharp electrode [47]). These simulations demonstrated that, if mature properties of g_pas_ were implemented in the simulated neuron, the GDP-induced [Cl^−^]_i_ changes were roughly comparable to the values observed in real CA3 pyramidal neurons (Figure 2g), while at g_pas_ typical for immature CA3 pyramidal neurons only marginal GDP-induced [Cl^−^]_i_ changes occurred.

In order to adapt the simulation of GDP-induced responses to the physiological properties of CA3 pyramidal neurons we incorporated an inward rectification in the background conductance (Appendix A). In addition, we had to increase the number of GABAergic synaptic inputs (n_GABA_) to compensate the influence of massive space clamp problems on the experimental determination of this parameter (Appendix A). For all further simulations in the reconstructed neurons we used the inward rectifying background conductance and implemented 302, 395, and 523 GABAergic synapses for P_HCO_3__ values of 0.0, 0.18, and 0.44, respectively. However, even with the inward rectifying conductance and 302 synaptic inputs the GDP-induced [Cl^−^]_i_ changes were smaller than observed under in-vitro conditions (Appendix A).

### 2.2. Influence of GABA Receptor Conductivity and Kinetics

Next we analyzed the influence of the GABAergic conductance (g_GABA_) on [Cl^−^]_i_ transients. Initial experiments in an isolated dendrite showed that initially the [Cl^−^]_i_ transient was localized underneath the synapse, and within 3 s a diffusional equilibration throughout the dendrite occurred (Appendix A). Therefore, we estimated the total amount of GABA-evoked [Cl^−^]_i_ changes by averaging the [Cl^−^]_i_ over all nodes of the dendrite 3 s after the GABAergic stimulus. To analyze the relation between total g_GABA_ and the [Cl^−^]_i_ changes, we first systematically increase g_GABA_ from 0.789 nS to 78.9 nS (Figure 3a). These simulations demonstrated that the GABA-evoked [Cl^−^]_i_ changes rose with increasing g_GABA_, but did not depend linearly on g_GABA_ (Figure 3b, black line). This nonlinear effect was due to the larger membrane depolarization upon stronger GABAergic stimulation, which reduced DF_Cl_ under this condition (data not shown). In an additional set of simulations, we enhanced the level of GABAergic stimulation by increasing the number of GABAergic synapses (n_GABA_) from 1 to 100, with g_GABA_ of 0.789 nS for each synapse. The synapses were for each n_GABA_ evenly distributed across the isolated dendrite. These simulations revealed that this distributed stimulation led to a reduced relative [Cl^−^]_i_ decrease at higher n_GABA_ (Figure 3b, red line), as compared to the previous simulation paradigm (Figure 3b, black line). This observation is most probably due to the fact that with distributed synapses E_m_ reaches more depolarized values close to E_Cl_ (−56.9 mV at 1 × 78.9 nS vs. −40.7 mV at 100 × 0.789 nS, data not shown).

To investigate whether a similar dependency between the amount of GABAergic inputs and [Cl^−^]_i_ could also be observed during a simulated GDP in a CA3 pyramidal neuron we increased g_GABA_ from 0.789 nS (Figure 3c, blue line) to 7.89 nS (red line) at each of the 302 synapses used to simulate a GDP. This 10× increase in g_GABA_ augmented the maximal GDP-induced [Cl^−^]_i_ decrease from 4.3 mM to 6.8 mM (Figure 3d). This surprisingly small effect was due to the fact that the increased g_GABA_ also reduced the average DF_Cl_ from −8.6 mV to −5 mV (Figure 3d). When a similar increase in the amount of GABAergic stimulation was implemented by a 10× increase in n_GABA_ (from 301 to 3010) a slightly larger maximal [Cl^−^]_i_ decrease by 7.1 mM was observed (Figure 3c,d, green line/symbols). This result indicates that the GDP-induced [Cl^−^]_i_ changes were close to saturation values when realistic values for n_GABA_, g_GABA_ and R_Input_ were implemented in a simulated CA3 pyramidal neuron.

In addition, we simulated how changes in the decay kinetics of GABA_A_ receptor-mediated currents (τ_GABA_) influence the [Cl^−^]_i_ transients (Appendix A). Systematic variation of τ_GABA_ between 10 ms and 1000 ms for a single synapse (g_GABA_ = 0.789 nS) in an isolated dendrite revealed that the average [Cl^−^]_i_ showed a nearly linear dependency on τ_GABA_ (Figure 4a, black line). If g_GABA_ was increased by a factor of 20 (g_GABA_ = 15.78 nS) the average [Cl^−^]_i_ concentration still showed a nearly linear dependency on τ_GABA_ (Figure 4a, red line). Since these responses suggested a strong influence of τ_GABA_ on the GABA-induced Cl^−^ fluxes, we also varied τ_GABA_ of all GABAergic synapses that were implemented on the reconstructed CA3 pyramidal neurons. These simulations revealed that an increase in τ_GABA_ indeed increased the GDP-induced [Cl^−^]_i_ changes (Figure 4b). The maximal GDP-induced decline in [Cl^−^]_i_ increases from 1.3 mM to 4.3 mM and 10.4 mM for τ_GABA_ of 3.7 ms, 37 ms and 370 ms, respectively (Figure 4c).

### 2.3. Contribution of the HCO_3_^−^ Conductance of GABA Receptors

In all previous experiments, we simulated GABA_A_ mediated responses under the simplified consideration that GABA_A_ receptors are ligand-gated Cl^−^ channels. However, GABA_A_ receptors are anion channels with a considerable HCO_3_^−^ permeability [1]. The relative HCO_3_^−^-permeability of GABA_A_ receptors (P_HCO_3__) ranges between 0.18 (determined in spinal cord neurons [48]) and 0.44 (determined in adult hippocampal neurons [49]), although also higher values have been suggested [1]. Therefore, we next simulated how P_HCO_3__ affects GABAergic E_m_ and [Cl^−^]_i_ responses upon stimulation of a single synapse in an isolated dendrite. Addition of a HCO_3_^−^ conductance to GABAergic currents induce a depolarizing shift in the peak depolarizations induced by GABAergic stimulation (Appendix A). Since this additional depolarization affected the DF_Cl_, the GABAergic [Cl^−^]_i_ changes were also influenced by P_HCO_3__. Under particular conditions, i.e. when E_m_ crossed E_Cl_ during synaptic responses, the GABAergic activation lead to biphasic [Cl^−^]_i_ changes (Figure 5a). For further analysis we plotted for such biphasic responses the maximal and minimal [Cl^−^]_i_ upon GABAergic stimulation (e.g. Figure 5b, blue lines). A systematic analysis of the effect of GABAergic inputs on the [Cl^−^]_i_ changes revealed that the [Cl^−^]_i_ changes were shifted towards more outward fluxes at higher P_HCO_3__ (Figure 5b), indicating that with increasing P_HCO_3__ a substantial [Cl^−^]_i_ increase is induced by GABAergic stimulation. 

However, these initial assumptions neglect the fact that the HCO_3_^−^ fluxes will also affect [HCO_3_^−^]_i_. Rapid regeneration of [HCO_3_^−^]_i_ levels by carbonic anhydrases, which stabilize [HCO_3_^−^]_i_, is absent in immature neurons [50]. Therefore, we first simulated the GABA-induced E_m_ and [Cl^−^]_i_ changes under the assumption that HCO_3_^−^ will not be replenished (by implementing a HCO_3_^−^ relaxation time constant (τ_HCO_3_^−^_) of 10 min) and is only redistributed by diffusion. These simulations revealed that the activation of GABA_A_ receptors induced a rapid decline in [HCO_3_^−^]_i_ (Appendix A). The [HCO_3_^−^]_i_ decline depended on both P_HCO_3__ and [Cl^−^]_i_ and was maximal at low [Cl^−^]_i_ with values of 1.8 mM and 2.3 mM for P_HCO_3__ of 0.18 and 0.44, respectively (Figure 5c). In line with this [HCO_3_^−^]_i_ decline, the GABAergic depolarization was drastically decreased (Figure 5d) under dynamic [HCO_3_^−^]_i_ conditions. The attenuation of activity-dependent [HCO_3_^−^] gradients also reduced the size of associated [Cl^−^]_i_ changes (Figure 5e) and at intermediate [Cl^−^]_i_ even reversed the effect (Appendix A). 

As suggested from the results in isolated dendrites, the GDP-induced depolarization simulated in the reconstructed neuron was augmented if P_HCO_3__ was increased from 0.0 to 0.18 and 0.44 under the assumption of stable [HCO_3_^−^] gradients (Figure 6a,c, shaded symbols). And because under these conditions E_m_ could become positive to E_Cl_, the Cl^−^ fluxes were enhanced and GDP-induced [Cl^−^]_i_ transients increased (Figure 6a,d, shaded symbols). Increasing P_HCO_3__ shifted the [Cl^−^]_i_ level at which GDP-induced [Cl^−^]_i_ transients change from influx to efflux, reflecting the impact of E_HCO_3__ on the DF_CL_. The maximal influence of P_HCO_3__ on the [Cl^−^]_i_ changes was observed at low [Cl^−^]_i_ (Figure 6d, shaded symbols), because at these conditions, the depolarizing effect of HCO_3_^−^ fluxes opposed the hyperpolarizing effects of E_Cl_. 

Implementation of a model that allowed dynamic [HCO_3_^−^]_i_ changes (using a τ_HCO_3__ of 10 min) in the reconstructed CA3 pyramidal neuron showed that the GDP-induced GABAergic currents induced massive changes in [HCO_3_^−^]_i_, depending on P_HCO_3__ (Appendix A). This GDP-induced [HCO_3_^−^]_i_ decrease during a GDP diminished the membrane depolarization (Figure 6b,c, plain lines/symbols), which in turn caused a drastic reduction in the GDP-induced [Cl^−^]_i_ transients (Figure 6b,d, plain lines/symbols). In summary, addition of P_HCO_3__ to GABAergic currents augmented the DF_Cl_ and thus the GDP-induced [Cl^−^]_i_ transients (Figure 6e). Using these parameters, the simulated GDP-induced [Cl^−^]_i_ transients resembled the size of [Cl^−^]_i_ transients observed in real cells, however, only in the quadrant with positive DF_Cl_ values (Figure 6e).

### 2.4. The Stability of HCO_3_^−^ Gradients Influences Activity-Dependent [Cl^−^]_i_ Transients 

The previous results clearly demonstrate that GABA_A_ receptor-mediated [HCO_3_^−^]_i_ transients massively influence the E_m_ and [Cl^−^]_i_ changes under these conditions. However, the two conditions used in these experiments (stable [HCO_3_^−^]_i_ or negligible [HCO_3_^−^]_i_ regeneration at τ_HCO_3__ of 10 min) are obviously not physiological in immature neurons, which lack carbonic anhydrases, but in which spontaneous CO_2_ hydration and/or transmembrane transport of HCO_3_^−^ can occur [50]. Therefore, we next investigated how τ_HCO_3__ influences the stability of [HCO_3_^−^] gradients and GABA induced [Cl^−^]_i_ transients. For that we systematically changed the decay-time of [HCO_3_^−^]_i_ relaxation (τ_HCO_3__) implemented in the NEURON model (Appendix A). A systematic simulation in isolated dendrites revealed that [Cl^−^]_i_ changes remained rather constant at τ_HCO_3__ ≥ 90 ms (Figure 7a). The half-maximal [Cl^−^]_i_ changes occurred at a τ_HCO_3__ around 10 ms, which is substantially shorter than the τ_HCO_3__ of ca. 70 ms for half-maximal [HCO_3_^−^]_i_ changes (Appendix A). More than 85% of the maximal [Cl^−^]_i_ changes took place at τ_HCO_3__ below 100 ms (Figure 7a). However, it must also be considered that a decreased temporal stability of the [HCO_3_^−^] gradient will also influence the lateral diffusion of HCO_3_^−^. Indeed, a systematic simulation of the spatial aspects of the activity-dependent [HCO_3_^−^]_i_ transients revealed that the stability of HCO_3_^−^ massively influenced the [HCO_3_^−^] gradient along the isolated dendrite (Figure 7b), although the maximal [HCO_3_^−^]_i_ change at the synaptic site was nearly saturated already at a τ_HCO_3__ ≥ 90 ms (Figure 7b). In accordance with the results obtained in isolated dendrites, also in the reconstructed CA3 pyramidal neuron τ_HCO_3__ had a large effect on the GDP-induced [HCO_3_^−^]_i_ transients (Appendix A), but only a minor effect on the associated [Cl^−^]_i_ changes (Figure 7c). For τ_HCO_3__ ≥ 90 ms the GDP-induced [Cl^−^]_i_ changes were only marginally affected (Figure 7c, Appendix A) by changes in τ_HCO_3__. At τ_HCO_3__ of 1 ms, the maximal GDP-induced [Cl^−^]_i_ decrease amounted to 5.2 mM, while it was 4.6 mM, 4.5 mM and 4.4 mM for τ_HCO_3__ values of 90 ms, 518 ms and 3 s, respectively. This small effect was also reflected by the minimal changes in the relation between DF_Cl_ and GDP-induced [Cl^−^]_i_ transients (Figure 7d).

GABAergic [51] and glutamatergic [52,53] synaptic transmission is accompanied by substantial pH changes. These pH changes, however, indirectly affect GABAergic transmission, since they alter the [HCO_3_^−^]_i_. To estimate, how such pH changes influence activity-dependent [Cl^−^]_i_ transients, we first simulated the effect of such pH shifts by constantly altering the pH value from 7.2 to 7.0 or 7.4 in an isolated dendrite. According to the Henderson-Hasselbalch equation, these pH shifts alter [HCO_3_^−^]_i_ from 14.1 mM to 9 mM or 22.7 mM, respectively. Because this pH-dependent differences in [HCO_3_^−^]_i_ affect DF_GABA_, the membrane depolarization upon GABA_A_ receptor activation was reduced at pH 7.0 and enhanced at a more alkaline pH of 7.4 (Figure 8a). In line with this altered GABAergic membrane depolarization, the DF_Cl_ during GABAergic stimulation was also affected, shifting the resulting Cl^−^ fluxes. This can be exemplified at intermediate [Cl^−^]_i_, where the biphasic Cl^−^ fluxes at a normal pH of 7.2, were transformed to Cl^−^ efflux at a pH of 7.0 and to a Cl^−^ influx at a pH of 7.4 (Figure 8a). A systematic analysis of Cl^−^ fluxes at different initial [Cl^−^]_i_ demonstrated that, in comparison to pH 7.2, the Cl^−^ influx at low initial [Cl^−^]_i_ was decreased at pH 7.0, while it was enhanced at pH 7.4 (Figure 8b). In contrast, the Cl^−^ efflux at high [Cl^−^]_i_ was enhanced at pH 7.0 and reduced at pH 7.4 (Figure 8d). In consequence, intracellular acidification shifted the [Cl^−^]_i_ range at which Cl^−^ efflux occurs to lower initial [Cl^−^]_i_, whereas intracellular alkalinization shifted this range to higher initial [Cl^−^]_i_ (Figure 8b).

Simulations in the reconstructed CA3 pyramidal neuron revealed that a lower pH of 7.0 led to smaller GDP-induced membrane depolarizations, as compared to the standard pH of 7.2 (Figure 8c, Appendix A, red line/symbols). This reduced depolarization resulted in a reduced GDP-associated Cl^−^ influx (4.6 mM vs. 7.6 mM) at low initial [Cl^−^]_i_ and in an increased Cl^−^ efflux (−6.1 mM vs. −4.3 mM) at high initial [Cl^−^]_i_ concentration (Figure 8d, red symbols). Conversely, at a higher pH of 7.4 the membrane responses during a GDP were more depolarized (Figure 8c, Appendix A green lines/symbols), which resulted in enhanced Cl^−^ influx of 11.2 mM at low initial [Cl^−^]_i_ and a decreased Cl^−^ efflux of −2.3 mM at low [Cl^−^]_i_ (Figure 8d, green symbols). In summary, these results demonstrate that the acidification associated with synaptic transmission reduced the activity-dependent [Cl^−^]_i_ transients at low [Cl^−^]_i_, while the activity-dependent Cl^−^ efflux at high [Cl^−^]_i_ was enhanced by such acidic shifts.

### 2.5. Influence of Transmembrane Cl^−^ Transport

Finally, we analyzed how the kinetics of the [Cl^−^]_i_ homeostasis influenced the temporal and spatial constrains of the activity-dependent [Cl^−^]_i_ changes. For that we systematically changed the decay-time of the [Cl^−^]_i_ relaxation (τ_Cl_) implemented in the NEURON model. Initial simulations in isolated and soma-attached dendrites revealed that for τ_Cl_ of ≥ 10 s the decay of [Cl^−^]_i_ response was dominated by diffusional exchange with the soma (Appendix A). At a τ_Cl_ of 321 s 99.2% of Cl^−^ fluxes were depleted by diffusional exchange with the soma, while at faster τ_Cl_ a substantial smaller fraction of only 83.5 % (τ_Cl_ = 10 s), 33.2 % (τ_Cl_ = 1 s) and 2.4% (τ_Cl_ = 100 ms) of the Cl^−^ fluxes was eliminated by diffusion from the dendrite to the soma. Analysis of the spatial distribution of [Cl^−^]_i_ along the dendrite revealed that τ_Cl_ also affects the size of activity-dependent [Cl^−^]_i_ changes at distant dendritic sites (Appendix A). The dominance of diffusional elimination of Cl^−^ was also reflected by the observation that at slow τ_Cl_ ≤ 10s the [Cl^−^]_i_ was substantial lower at the proximal than at the distal end of the dendrite (Appendix A). 

To analyze the influence of τ_Cl_ on the spatial aspects of the [Cl^−^]_i_ transients we implemented two simultaneous GABAergic inputs that were located equidistant to the [Cl^−^]_i_ recording site at distances of 10 µm, 30 µm, 100 µm and 300 µm and systematically increased τ_Cl_ from 1 ms to 220 s (Figure 9a). These simulations revealed not only that the maximal [Cl^−^]_i_ depended on the distance between GABAergic stimulation sites and the node of [Cl^−^]_i_ determination, but also that τ_Cl_ critically influenced the [Cl^−^]_i_ change at a given distance to the stimulation sites (Figure 9a). This dependency between spatial restrictions of activity-dependent [Cl^−^]_i_ changes and τ_Cl_ was quantified by the τ_Cl_ at which half-maximal [Cl^−^]_i_ changes occur (τ^Cl^_50_). If the distance of the GABAergic synapses was 10 µm τ^Cl^_50_ amounted to 12 ms, and this τ^Cl^_50_ increased to 60.5 ms, 726 ms and 4.6 s at synaptic distances of 30 µm, 100 µm and 300 µm, respectively. To analyze the temporal aspects of [Cl^−^]_i_ summation we simulated five consecutive GABA stimulations at frequencies (f_GABA_) of 0.3 Hz, 1 Hz, 3 Hz and 10 Hz and determine the [Cl^−^]_i_ at the stimulation site, while systematically varying τ_Cl_ (Figure 9b). These simulations revealed a sigmoidal dependency between τ_Cl_ and the temporal summation of [Cl^−^]_i_. A larger amount of [Cl^−^]_i_ summation and a lower τ^Cl^_50_ was observed at higher frequencies. The τ^Cl^_50_ amounted to 1.9 s for f_GABA_ of 0.1 Hz, 931 ms for f_GABA_ of 1 Hz, 268 ms for f_GABA_ of 3 Hz, and 53 ms for f_GABA_ of 10 Hz. In summary, these results demonstrated that τ_Cl_ values of less than 1 s are required to prevent substantial activity-dependent [Cl^−^]_i_ changes in the spatial and/or temporal domain at f_GABA_ ≥ 1Hz and less than 100 µm distance between synaptic sites.

In accordance with these results in single dendrites, all previous simulations of GDP-induced [Cl^−^]_i_ transients in the reconstructed CA3 pyramidal cells revealed substantial [Cl^−^]_i_ changes, because in these simulations the experimentally determined τ_Cl_ of 174 s for NKCC1-mediated active Cl^−^ re-accumulation and of 321 s for passive Cl^−^ reduction were implemented and during a GDP stimulation a high frequency of GABAergic input was applied. In order to get more insights into how the capacity of [Cl^−^]_i_ regulation systems can influence activity-dependent [Cl^−^]_i_ transients within a realistic dendritic compartment, we finally simulated how different τ_Cl_ influenced the GDP-induced [Cl^−^]_i_ transients in the reconstructed CA3 neuron (Figure 9c,d). This simulation revealed that decreasing τ_Cl_ from the experimentally determined values >100 s to 10 s or 1 s had only a minimal impact of the GDP-induced [Cl^−^]_i_ transients (Figure 9c). The maximal GDP-induced [Cl^−^]_i_ change amounted to 4.41 mM at a τ_Cl_ of 10 s and to 4.21 mM at a τ_Cl_ of 1 s, but were reduced to 2.99 mM at a τ_Cl_ of 0.1 s and to 0.88 mM at a τ_Cl_ of 10 ms. These results demonstrate that fast and efficient [Cl^−^]_i_ homeostatic processes are required to limit GDP-induced [Cl^−^]_i_ transients. Accordingly, the Δ[Cl^−^]_i_ vs. DF_Cl_ plot also revealed comparable GDP-induced [Cl^−^]_i_ changes at τ_Cl_ of 10 s and 1 s, and smaller [Cl^−^]_i_ changes at a τ_Cl_ of 100 ms (Figure 9d). Only a further reduction in τ_Cl_ to 10 ms substantially suppressed GDP-induced [Cl^−^]_i_ changes. In summary, these results indicate that τ_Cl_ influences the temporal and spatial properties of activity-dependent [Cl^−^]_i_ changes, but that τ_Cl_ values that are substantially smaller than the experimentally determined values are required to suppress activity-dependent [Cl^−^]_i_ changes.

## 3. Discussion

In the present study we used a detailed biophysical compartmental modeling in the NEURON environment to systematically investigate how several cellular and molecular neuronal parameters influence the GABA_A_ receptor-mediated [Cl^−^]_i_ changes. The main observations of this study can be summarized as follows: (i) A high R_input_ reduces activity-dependent [Cl^−^]_i_ transients, while at low R_input_ considerable activity-dependent [Cl^−^]_i_ transients can be observed. (ii) The activity-dependent [Cl^−^]_i_ transients show a logarithmic impact of g_GABA_, while τ_GABA_ has in a wide g_GABA_ range a nearly linear influence on [Cl^−^]_i_. (iii) The P_HCO_3__ of GABA_A_-receptors enhances activity-dependent [Cl^−^]_i_ transients, but with instable [HCO_3_^−^] gradients this effect is largely diminished. (iv) Activity-dependent Cl^−^ fluxes where shifted toward efflux at acidic and towards influx at alkaline pH. (v) τ_Cl_ has a major impact on the spatiotemporal aspects of activity-dependent [Cl^−^]_i_ transients, but unrealistically fast τ_Cl_ values are required to prevent [Cl^−^]_i_ transients at physiologically-relevant activity levels.

By 1990, it was suggested by Qian and Sejnowski [54] that the Cl^−^ fluxes via activated GABA_A_ receptors will dissipate the Cl^−^ gradient in small compartments and thus mediate potentially instable inhibitory responses. This theoretical assumption was proven by experimental studies, which demonstrated that massive GABAergic activation can shift hyperpolarizing responses toward depolarization [12,21] and induce [Cl^−^]_i_ transients [55]. In the past the physiological and pathophysiological consequences of such activity-dependent [Cl^−^]_i_ changes have been investigated and discussed [13,17,20,36,56] and the basic principles of activity-dependent [Cl^−^]_i_ changes and their implications for neuronal information processing have been modeled [7,15,16,22,57,58]. However, the complex interplay and contribution of passive membrane leak, GABA_A_ conductance, Cl^−^ diffusion/ transport and stability of [HCO_3_^−^] gradients to these activity-dependent [Cl^−^]_i_ changes have not yet been systematically investigated.

Our simulations revealed a strong dependence between R_Input_ and the GABA_A_ receptor induced [Cl^−^]_i_ transients. While at high R_Input_ GABA-induced [Cl^−^]_i_ changes were minimal, they increased in a nonlinear relation with decreasing R_Input_ (Figure 1c). This relation between R_Input_ and the [Cl^−^]_i_ changes is due to the fact that at high R_Input_ even small GABAergic currents bring E_m_ close to E_Cl_, which minimizes DF_Cl_ and thus the Cl^−^ fluxes (Figure 1c). At low R_Input_ the passive membrane conductance stabilizes E_m_ and thus DF_Cl_. In consequence, larger Cl^−^ fluxes can be expected. Accordingly, implementation of “adult like” membrane properties [47] in a reconstructed immature neuron massively enhanced activity-dependent [Cl^−^]_i_ changes (Figure 1c). In contrast, it seems obvious that immature neurons, with their high R_Input_ [59], are less susceptible to activity-dependent [Cl^−^]_i_ changes. 

However, in this respect, it is important to consider that in immature neurons [Cl^−^]_i_ is high and GABAergic responses are depolarizing [30,60,61], therefore activity-dependent Cl^−^ fluxes are directed outward and are leading to [Cl^−^]_i_ decrease [42,44]. In addition, the HCO_3_^−^-permeability of GABA_A_ receptors also needs to be considered. The high R_Input_ in immature neurons causes E_m_ to approach E_GABA_, which normally is positive to E_Cl_ due to the HCO_3_^−^-permeability of GABA_A_-receptors [1,4]. Therefore, stable Cl^−^ influx would be expected under this condition and [Cl^−^]_i_ should ultimately approach the value defined by E_HCO_3__, which at steady-state HCO_3_^−^ gradients ([HCO_3_^−^]_i_ = 14.1 mM and [HCO_3_^−^]_e_ = 26 mM) amounts to 72.4 mM. This estimation suggests that under certain conditions GABAergic activity can even increase [Cl^−^]_i_ from the already high [Cl^−^]_i_-levels in immature neurons (see Appendix A). But further properties of activity-dependent [Cl^−^]_i_ changes observed in our simulations protect immature neurons from excessive [Cl^−^]_i_ increases. In particular, we found that the influence of P_HCO_3__ is relatively small at high [Cl^−^]_i_ (Figure 6d), due to the fact that under this condition the contribution of E_HCO_3__ to E_GABA_ is small (as described by the Goldman-Hodgkin-Katz-Equation [1]). In addition, in immature neurons the HCO_3_^−^ gradient is instable because they lack carbonic anhydrases [50], which additionally attenuates the depolarizing effect of HCO_3_^−^ on E_m_ and thus reduces DF_Cl_.

Under the assumption of a stable HCO_3_^−^ gradient, E_GABA_ is in a wide range positive to E_Cl_, thereby permitting Cl^−^ influx during GABAergic stimulation, unless the thermodynamics equilibrium at 72.4 mM is reached. This is supported by the observation that no stable steady-state [Cl^−^]_i_ is reached with realistic P_HCO_3_^−^_ values of 0.18 or 0.44 in our simulation (Appendix A, shaded lines). Experimental studies indeed demonstrated that massive GABAergic stimulation can shift E_GABA_ from hyperpolarizing towards depolarizing and even excitation [12,62,63,64]. On the other hand, if we implemented in our model that [HCO_3_^−^]_i_ can be altered by GABA_A_ receptors, the activity-dependent [Cl^−^]_i_ changes were massively reduced (Figure 5e). Our simulations also demonstrate that the switch from static [HCO_3_^−^] to dynamic [HCO_3_^−^] condition shifts the [Cl^−^]_i_ setpoint at which activity-dependent Cl^−^ influx was replaced by Cl^−^ efflux to considerably lower values (Figure 5e). This observation is due to the fact that the depolarizing HCO_3_^−^ fluxes through the GABA_A_ receptor are attenuated by the dissipating HCO_3_^−^ gradient, which reduces E_GABA_ and thus DF_Cl_. Similar conclusions were drawn from experiments in which a block of carbonic anhydrases with acetazolamide, which provides a pharmacological destabilization of [HCO_3_^−^], also reduces E_GABA_ shifts ([18], but see [62]). We conclude from these observations that the lack of carbonic anhydrase VII in immature neurons [50,65] may serve to limit the activity-dependent [Cl^−^]_i_ changes in these neurons. 

On the other hand, our simulations in a reconstructed CA3 pyramidal neuron revealed that although massive [HCO_3_^−^]_i_ changes are induced under this condition, the total GDP-induced [Cl^−^]_i_ change was only marginally affected by variations in τ_HCO_3__ (Figure 7d). This lack of effect was most probably due to the fact that the activity-dependent [HCO_3_^−^]_i_ change was already maximal at a τ_HCO_3__ of 90 ms at the synaptic site. This local saturated [HCO_3_^−^]_i_ change at the subsynaptic site is the only determinant for the synaptic effects of the [HCO_3_^−^]_i_. Our simulations also suggest that for an effective, physiologically relevant control of [HCO_3_^−^]_i_ during GABAergic activity τ_HCO_3__ should be less than 70 ms (Appendix A). This fast relaxation time requires fast molecular processes that allow effective elimination of HCO_3_^−^. Indeed, carbonic anhydrases, the enzymes that mediate the degradation or regeneration of HCO_3_^−^ into/from H_2_O and CO_2_, are among the fastest enzymes known. The k_cat_ of murine carbonic anhydrase VII for the hydration oh CO_2_ is 4.5 × 10^5^ s^−1^ at physiological pH [66]. From the assumption that the ca. 2mM [HCO_3_^−^]_i_ change in the dendrite corresponds at a dendritic volume of 16 pL to ca. 0.3 fmol HCO_3_^−^ ions, it can be estimated that about 4000 molecules of carbonic anhydrase VII are required to replenish the lost HCO_3_^−^ within 100 ms. This estimation suggests that it is reasonable that sufficient carbonic anhydrase activity can be located in the dendritic compartment to reliably stabilize [HCO_3_^−^]_i_. However, as the reaction mediated by carbonic anhydrases includes H^+^ ions, the kinetics and thermodynamics of this process depends on the intracellular pH [67]. Thus dendritic H^+^-buffering and handling indirectly also affects activity-dependent [Cl^−^]_i_ changes [15]. The acidification associated with neuronal activity [52,53] will slow down the kinetics of carbonic anhydrases [66]. However, this effect is negligible in comparison to the effect of the intracellular pH on [HCO_3_^−^]_i_. The intracellular pH is an essential parameter that determines [HCO_3_^−^]_i_ [67]. Thus the intracellular acidification observed upon activation of GABAergic and glutamatergic synapses [51,52,53] will alter [HCO_3_^−^]_i_ and subsequently influence GABAergic transmission. Our simulation revealed that an intracellular acidification will reduce the activity-dependent [Cl^−^]_i_ changes at low [Cl^−^]_i_. This result, which is in accordance with a previous simulation [15], indicate that in adult neurons a parallel acidification will limit Cl^−^ influx and thus stabilize inhibitory transmission In contrast, at a high [Cl^−^]_i_ typical for immature neuron the intracellular acidification enhanced the activity-dependent Cl^−^ efflux and may contribute to the loss of depolarizing drive and putative excitatory effects after strong GABAergic stimulation.

Another factor that has a stringent effect on activity-dependent [Cl^−^]_i_ changes in our simulations is τ_GABA._ This confirmed and extended previous computational analyses (c.f. Figure 4d in [7]). It has been found that in general the decay kinetics of GABAergic transmission get faster during development [68]. Therefore the slow decay kinetics of GABAergic transmission in immature neurons [68,69] may be a factor that enables activity-dependent [Cl^−^]_i_ transients, while the faster GABAergic postsynaptic currents in mature neurons not only improve the temporal precision of GABAergic transmission [68], but also the stability of inhibition. While a stable inhibition is a prerequisite for the proper function of mature neuronal networks, dynamic changes in [Cl^−^]_i_ can be mandatory for physiological relevant functional features of the immature central nervous system. It has been suggested that activity-dependent changes in [Cl^−^]_i_, and the resulting switch from GABAergic inhibition to excitation, can underlie oscillatory activity [13]. In addition, in the immature nervous system the resting [Cl^−^]_i_ is decreased by GABAergic activity, which will result in a diminished excitatory drive and/or a dominance of shunting inhibition and may thus serve to limit a possible excitatory effect of GABA [40]. Therefore, for immature neurons an unstable [Cl^−^]_i_ homeostasis may be functionally relevant, as it allows activity-dependent scaling of [Cl^−^]_i_-dependent synaptic transmission [42,43].

In consequence, the molecular configuration of immature neurons (high [Cl^−^]_i_, long τ_GABA_ and missing CA-VII) will generate conditions that allow limited activity-dependent [Cl^−^]_i_ decreases. This, in addition to the aforementioned effect of the high input resistance, may be an explanation why the [Cl^−^]_i_ homeostasis of immature neurons is maintained by a relatively ineffective transmembrane Cl^−^ transport [29]. In contrast, in mature neurons the situation is different. In the adult brain, the low [Cl^−^]_i_ is needed to maintain hyperpolarizing inhibition [1] and an activity-dependent [Cl^−^]_i_ increase will attenuate membrane hyperpolarization. While it is obvious that massive changes in [Cl^−^]_i_ will impair GABAergic inhibition and can lead to hyperexcitability [18], recent modeling experiments demonstrate that even minimal changes in the capacity of Cl^−^-extrusion can have strong effects on information processing and storage in neurons [22]. Although the low R_Input_ and the fast τ_GABA_ counteracts activity-dependent [Cl^−^]_i_ increase in adult neurons, their low [Cl^−^]_i_ and their effective carbonic anhydrases can lead to substantial [Cl^−^]_i_ changes in their dendrite. The adverse effect of such local activity-dependent [Cl^−^]_i_ increases is enhanced by the more elaborated dendritic compartment in mature neurons, which limits diffusional elimination of Cl^−^-ions [13].

Our simulations reveal that the connection of an isolated dendrite to the soma drastically reduces the equilibrium [Cl^−^]_i_ after synaptic stimulation (Appendix A), demonstrating the important role of diffusional Cl^−^ elimination under this condition. The large volume to surface ratio (and thus volume to conductance ratio) of the soma enables this compartment to serve as Cl^−^ sink in these in-silico experiments. Also in-vitro it has been demonstrated that activation of dendritic GABA_A_ receptors induced massive shifts in E_GABA_, whereas only minimal changes were observed upon perisomatic stimulation [10,12,26]. The dominance of perisomatic GABAergic terminals [70] may be related to the requirement of stable [Cl^−^] gradients to maintain stable inhibition over a wide range of activity levels. However, the diffusion of [Cl^−^]_i_ through dendrite is a relatively slow and inefficient process, due to the small diameter in distal dendrites [7,55]. Addition of spines to dendrites drastically slow down diffusion along dendrites [16], suggesting that the complexity of the dendritic compartment (i.e. the number of arborizations that enhance tortuosity in the dendritic compartment) hinders Cl^−^-elimination by diffusion to the soma. Therefore, active elimination of Cl^−^ from the cytoplasm is required to prevent or minimize activity-dependent [Cl^−^]_i_ changes in the elaborated dendritic compartment of adult neurons. 

The elementary role of transmembrane Cl^−^ transporters for neuronal [Cl^−^]_i_ homeostasis has been shown by a variety of studies [2,29,32,58,71]. Modeling studies revealed that slightly altered rates of transmembrane Cl^−^-transport, which does only marginally affect resting [Cl^−^]_i_ levels, have a strong effect on the spatiotemporal distribution of activity-dependent [Cl^−^]_i_-transients in dendrites [15]. Therefore it is not surprising that the simulation of an enhanced capacity of transmembrane Cl^−^ transport by increasing τ_Cl_ attenuates activity-dependent [Cl^−^]_i_ transients. However, to minimize these [Cl^−^]_i_ transients a rather low τ_Cl_ of < 100 ms is required. These low τ_Cl_ values are several orders of magnitude below the experimentally determined kinetics of the NKCC1-mediated Cl^−^-accumulation (τ_Cl_ = 158 s) in immature neurons [29]. Because of this slow kinetic of transmembrane transport of Cl^−^ in immature neurons, we also consider that a Cl^−^/HCO_3_^−^ exchange mediated by the anion exchanger in immature neurons [72] has only a marginal effect on both activity-dependent [Cl^−^]_i_ and [HCO_3_^−^]_i_ transients in these neurons. In the mature situation (low [Cl^−^]_i_ and effective transmembrane [Cl^−^]_i_ transport), modeling studies suggest that an interference between [Cl^−^]_i_ and [HCO_3_^−^]_i_ by this mechanism can reduce the activity-dependent [Cl^−^]_i_ changes [15].

While it is generally assumed that the neuron-specific Cl^−^-extruder KCC2 mediates more efficient Cl^−^ transport than NKCC1, only few experimental studies addressed the kinetics of KCC2-dependent Cl^−^-extrusion. Experiments in brain stem neurons demonstrated that KCC2 mediated Cl^−^-extrusion after [Cl^−^]_i_ increase by ca. 10 mM requires several minutes [73]. In contrast, in-vivo experiments revealed that the activity-dependent [Cl^−^]_i_ increase after an epileptic seizure recovered within less than 30 s [9] and in hippocampal slices GABA-induced [Cl^−^]_i_ transients recovered back to low steady-stale levels with a time constant of 3.3 s [12]. However, it is not clear how diffusional processes and/or the kinetics of the used Cl^−^ sensor contribute to these kinetic properties. Simulations suggest that with realistic KCC2 levels τ_Cl_ in the distal dendrites (≥ 200 µm from the soma) is between 100 ms and 200 ms [15], and thus probably lower than estimated from experimental data. Even this time constant is higher than the τ_Cl_ required in our simulations to prevent local activity-dependent [Cl^−^]_i_ changes, suggesting that considerable [Cl^−^]_i_ changes can occur at GABAergic synaptic sites. While our and other simulations [15] suggest these transients may be restricted to local dendritic domains, it must be emphasized that subtle changes in the efficacy of KCC2 mediated Cl^−^-transport can already enhance the excitability in single neurons because activity-dependent [Cl^−^]_i_ transients may superimpose these effects [22]. In consequence, impairments of KCC2 mediated Cl^−^ transport can led to a breakdown of sufficient inhibition in neuronal networks and contribute to hyperexcitability [15,17,20,56,74]. In this respect it is also relevant to consider that the activity of both NKCC1 and KCC2 are regulated by a variety of processes [75,76,77,78]. This indicates that the spatiotemporal [Cl^−^]_i_ dynamics in the dendritic compartment may be adapted to the functional states. 

The limitation of our model to fully describe GDP-induced [Cl^−^]_i_ transients in CA3 pyramidal neurons is obvious from the fact, that we massively underestimate the [Cl^−^]_i_ decrease observed in real CA3 pyramidal neurons at high [Cl^−^]_i_ (e.g. Figure 6e). Therefore, additional factors must be proposed, which enhance the GABA_A_-receptor-mediated Cl^−^ efflux. Possible mechanisms that improve Cl^−^ efflux are e.g. an inhomogeneous distribution of voltage-activated K^+^ channels in the dendritic compartment, an underestimation of n_GABA_ in our in-vitro experiments due to voltage-clamp errors in the elaborated dendrite [79], or the effect of glutamatergic transmission during a GDP [20,80]. In addition, we also found that τ_GABA_ has a major impact on the GDP-induced [Cl^−^]_i_ transients, and it might well be that the decay kinetics of spontaneous GABAergic PSCs of 37 ms [45] reflect the kinetic properties of a subpopulation of GABAergic inputs, that is less involved in the generation of GDPs. Finally, in our simulations the dendrite was implemented as a hollow tube with a diameter determined from the histological reconstruction. Under realistic assumptions the neuron is, however, filled with cytoplasm that contains large proteins, particles of different sizes and vesicles and tubes of intracellular organelles. Thus the free, “unexcluded” volume in the cytoplasm is restricted to an estimated fraction of ca. 60%, a principle termed cytoplasmic crowding [81]. This restricted free cytoplasmic water volume will increase the size of [Cl^−^]_i_ transients upon identical Cl^−^ fluxes.

## 4. Materials and Methods 

### Compartmental Modeling

The biophysically realistic compartmental modelling was performed using the NEURON environment (neuron.yale.edu). The source code of models and stimulation files used in the present paper can be found in ModelDB [82] at http://modeldb.yale.edu/253369 (access date 14 March 2019) and was included in the Appendix A of this publication. For compartmental modelling we used either a simple ball and stick model (soma with d = 20 µm, linear dendrite with l = 200 µm and 103 nodes) or a reconstructed CA3 pyramidal cell (from Lombardi et al. [45]). Except where noted the dendrite was detached from the soma to analyse dendritic [Cl^−^]_i_ transients. The reconstructed neuron resembled the somatodendritic morphology of a typical immature CA3 pyramidal neuron (see Figure 2a,b). For this purpose images of a biocytin-filled neuron [83,84] were taken with 60× oil-immersion objectives and the somatodendritic morphology was reconstructed using Fiji (www.fiji.sc). It contained a soma (d = 15 µm), a dendritic trunk (d = 2 µm, l = 32 µm, 9 segments) and 56 dendrites (d = 0.36 µm, 9 segments each). In all of these compartments a specific axial resistance (*R_a_*) of 34.5 Ωcm and a specific membrane capacitance (*C_m_*) of 1 µF cm^−2^; were implemented. The specific membrane conductance (g*_pas_*) varied (see Figure 1a and Figure 2e) and in the majority of the experiments was modeled by a voltage dependent process given by a Boltzmann-like equation: (1)$$g_{\mathrm{pas}}=g_{\min}+\frac{g_{\max}}{\left(1+\exp\left(\frac{\left(E_{m}-E_{50}\right)}{s}\right)\right)}$$
with g_max_ = 0.002800 S/cm^2^ (experimentally determined g_Input_ at depolarized potentials, see Appendix A), gmin = 0.000660 S/cm^2^ (experimentally determined minimal g_Input_ at hyperpolarized potentials, see Figure 3a), e50 = −31 mV (half-maximal voltage), s = −6 (slope of the voltage-dependency). The reversal potential of this voltage-depended g_pas_ was set to −60 mV.

GABA_A_ synapses were simulated as a postsynaptic parallel Cl^−^ and HCO_3_^−^ conductance with exponential rise and exponential decay [7]:
I_GABA_ = I_Cl_ + I_HCO_3__ = 1/(1+P) g_GABA_ (V–E_Cl_) + P/(1+P) g_GABA_ (V–E_HCO_3__)(2)
where P is a fractional ionic conductance that was used to split the GABA_A_ conductance (g_GABA_) into Cl^−^ and HCO_3_^−^ conductance. E_Cl_ and E_HCO_3__ were calculated from Nernst equation. The GABA_A_ conductance was modeled using a two-term exponential function, using separate values of rise time (0.5 ms) and decay time (variable, mostly 37 ms) [45]. Parameters used in our simulations were as follows: [Cl^−^]_o_ = 133.5 mM, [HCO_3_^−^]_i_ = 14.1 mM, [HCO_3_^−^]_o_ = 24 mM, temperature = 31 °C, P = 0.44 [49]. For the ball and stick model a single GABA_A_ synapse was placed in the middle of the dendrite, except where noted. For the simulation of a GDP in the reconstructed CA3 neuron 101–3020 GABAergic synapses were randomly distributed within the dendrites of the reconstructed neuron. GABA inputs were activated stochastically using a normal distribution (µ = 600ms, σ = 900 ms) that emulates the distribution of GABAergic PSCs during a GDP observed in immature hippocampal CA3 pyramidal neurons [45]. The properties of these synapses were always given in the results part and/or the corresponding figure legends. 

From the quotient between the charge transfer of a GDP and of spontaneous GABAergic postsynaptic currents at a holding potential (V_Hold_) of 0 mV it was estimated that 101 GABAergic inputs underlie a GDP [45]. To compensate for the space-clamp problems during a GDP, that were not considered by Lombardi et al. [45], we simulated the charge transfer during a GDP under their experimental conditions ([Cl^−^]_i_ = 10 mM, V_Hold_ = 0 mV) and determined that 302, 395, and 523 (for P_HCO_3__ values of 0.0, 0.18, and 0.44, respectively) GABAergic synapses are required to generate the observed GDP-induced charge transfer (Appendix A). For these experiments we implement the single-electrode voltage clamp procedure provided by NEURON, using an access resistance of 5 MΩ. The charge transfer was calculated from the integral of the holding currents (I_Hold_) during the GDP.

For the modeling of the GABA_A_ receptor-induced [Cl^−^]_i_ and [HCO_3_^−^]_i_ changes, we calculated ion diffusion and uptake by standard compartmental diffusion modeling [16,85,86,87]. To simulate intracellular Cl^−^ and HCO_3_^−^ dynamics, we adapted our previously published model [7]. Longitudinal Cl^−^ and HCO_3_^−^ diffusion along dendrites was modeled as the exchange of anions between adjacent compartments. For radial diffusion, the volume was discretized into a series of 4 concentric shells around a cylindrical core [85] and Cl^−^ or HCO_3_^−^ was allowed to flow between adjacent shells [88]. The free diffusion coefficient of Cl^−^ inside neurons was set to 2 µm^2^/ms [55,89]. Since the cytoplasmatic diffusion constant for HCO_3_^−^ is, to our knowledge, unknown, we also used a value of 2 µm^2^/ms. To simulate transmembrane transport of Cl^−^ and HCO_3_^−^, we implemented an exponential relaxation process for [Cl^−^]_i_ and [HCO_3_^−^]_i_ to resting levels [Cl^−^]_i_^rest^ or [HCO_3_^−^]_i_^rest^ with a time constant τ_Ion_.
(3)$$\frac{d{\left[{\mathrm{Ion}}^{-}\right]}_{i}}{\mathrm{dt}}=\frac{{\left[{\mathrm{Ion}}^{-}\right]}_{i}^{\mathrm{rest}}-e{\left[{\mathrm{Ion}}^{-}\right]}_{i}}{\tau_{\mathrm{Ion}}}$$
Cl^−^ transport was in most experiments (if not otherwise noted) modeled as bimodal process, for [Cl^−^]_i_ < [Cl^−^]_i_^rest^ τ_Ion_ was set to 174 s to emulate an NKCC1-like Cl^−^ transport mechanism. For [Cl^−^]_i_ > [Cl^−^]_i_^rest^ τ_Ion_ was set to 321 s to emulate passive Cl^−^ efflux (both values obtained from unpublished experiments on immature rat CA3 hippocampal neurons). 

The impact of GABAergic Cl^−^ currents on [Cl^−^]_i_ and [HCO_3_^−^]_i_ was calculated as:(4)$$\frac{d{\left[{\mathrm{Ion}}^{-}\right]}_{i}}{\mathrm{dt}}=\frac{1}{F}\frac{I_{\mathrm{Ion}}}{\mathrm{volume}}$$

To simulate the GABAergic activity during a GDP, a unitary peak conductance of 0.789 nS and a decay of 37 ms were applied to each GABAergic synapse. These values resulted in a unitary currents of pA, which was in accordance with the mean amplitude of spontaneous GABAergic postsynaptic currents in CA3 paramidal neurons [45].

For the isolated neurons the [Cl^−^]_i_ and [HCO_3_^−^]_i_ concentration was averaged over all segments of the dendrite, except where noted. For the simulated neurons we analyzed mean [Cl^−^]_i_ and [HCO_3_^−^]_i_ of all dendrites: (5)$${\left[Cl^{-}\right]}_{i}=\frac{1}{n_{dend}}\times \overset{n_{dend}}{\underset{j=1}{\sum }}{\left[Cl^{-}\right]}_{i}^{Dend\left(j\right)\;@\;0.5\;of\;total\;length}$$

This procedure mimics the experimental procedure of Lombardi et al [45], who determined E_GABA_ by focal application in the dendritic compartment. 

For the calculation of Δ[Cl^−^]_i_ the maximal deviation of [Cl^−^]_i_ upon a GABAergic stimulus ([Cl^−^]_i_^S^) was subtracted from the resting [Cl^−^]_i_ before the stimulus ([Cl^−^]_i_^R^). For biphasic responses both minimal and maximal [Cl^−^]_i_^R^ were determined and Δ[Cl^−^]_i_ was calculated as:(6)$$\Delta {\left[Cl^{-}\right]}_{i}=\;{\left[Cl^{-}\right]}_{i}^{S,\;\min}-{\left[Cl^{-}\right]}_{i}^{R}\;if\;abs\left({\left[Cl^{-}\right]}_{i}^{S,\;\min}\right)>abs\left({\left[Cl^{-}\right]}_{i}^{S,\;\max}\right)$$
(7)$$\Delta {\left[Cl^{-}\right]}_{i}={\left[Cl^{-}\right]}_{i}^{S,\;\max}-{\left[Cl^{-}\right]}_{i}^{R}\;if\;abs\left({\left[Cl^{-}\right]}_{i}^{S,\;\min}\right)\leq \;abs\left({\left[Cl^{-}\right]}_{i}^{S,\;\max}\right)$$

The driving-force of Cl^−^ (DF_Cl_) was calculated from the difference between the average E_m_ during a GDP and E_Cl_ (DF_Cl_ = E_m_ – E_Cl_). To calculate the ratio between transmembrane [Cl^−^]_i_ transport and diffusional [Cl^−^]_i_ depletion into the soma, we normalized the diffusional exchange between the last somatic node and the soma (as calculated from Fick’s law) to conditions were transmembrane [Cl^−^]_i_ loss was absent (τ_Cl_ = 10^9^ ms) and diffusional dendrite to soma transport was allowed to equilibrate for 2 min.

All electrophysiological data were taken from our previous publication [45]. However, for a comparison of these results with the simulations, we had to take different P_HCO_3__ into account. Therefore the GDP-induced [Cl^−^]_i_ changes were recalculated using P_HCO_3__ values of 0.0, 0.18 (determined in spinal cord neurons [48]) and 0.44 (determined in adult hippocampal neurons [49]). The [Cl^−^]_i_ was calculated from E_GABA_ with the Goldman-Hodgkin-Katz equation:(8)$$E_{\mathrm{GGABA}}=\frac{\mathrm{RT}}{\mathrm{ZF}}\times \ln\left(\frac{P_{Cl}{\left[Cl^{-}\right]}_{e}+P_{{\mathrm{HCO}}_{3}}{\left[{\mathrm{HCO}}_{3}{}^{-}\right]}_{e}}{P_{Cl}{\left[Cl^{-}\right]}_{i}+P_{{\mathrm{HCO}}_{3}}{\left[{\mathrm{HCO}}_{3}{}^{-}\right]}_{i}}\right)$$

For the calculation of [Cl^−^]_i_ from E_GABA_ we used a [Cl^−^]_e_ of 133.5 mM, an extracellular HCO_3_^−^ concentration ([HCO_3_^−^]_e_) of 24 mM and assumed a constant [HCO_3_^−^]_i_ of 14.1 mM (calculated from an intracellular pH of 7.2 [90], a CO_2_ pressure (pCO_2_) of 38 mmHg, a Henry coefficient (α) of 0.0318 mM/mmHg and a pKs of 6.128 [91] with the Henderson-Hasselbalch equation), if not otherwise mentioned.
(9)$${\left[{\mathrm{HCO}}_{3}{}^{-}\right]}_{i}=10^{\left(\mathrm{pH}-\mathrm{pKs}+\log\left(\alpha \times {\mathrm{pCO}}_{2}\right)\right)}$$

R_Input_ was calculated from the E_m_ response upon a simulated current injection (I_Inj_) according to Ohms law:(10)$$R_{\mathrm{Input}}=\frac{E_{m}}{I_{Inj}}$$

## Figures and Tables

**Figure 1 ijms-20-01416-f001:**
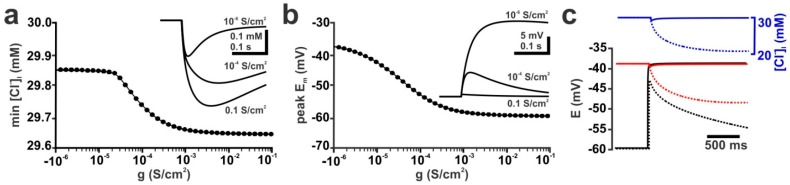
Passive membrane conductance (g_pas_) influences GABA-induced [Cl^−^]_i_ transients. At low g_pas_ values, GABAergic currents induce strong depolarization, attenuating the driving force for Cl^−^ ions and thereby decreasing Cl^−^ fluxes. (**a**) The [Cl^−^]_i_ transients induced by a single GABAergic stimulation (g = 0.789 nS, τ = 37 ms, P_HCO_3__ = 0, [Cl^−^]_i_ = 30 mM) show a strong dependency on g_pas_. Three typical traces are displayed as inset. (**b**) The GABA-induced membrane depolarization also shows a sigmoidal dependency on g_pas_. (**c**) Effect of g_pas_ on E_m_ (black lines), E_Cl_ (red lines) and [Cl^−^]_i_ (blue lines) in an isolated dendrite using constant GABAergic currents (g_GABA_ = 0.1 µS). Note that at low g_pas_ values (0.1 nS/cm^2^, solid lines) E_m_ approximates E_Cl_, while at high g_pas_ (18 mS/cm^2^, dashed lines) E_m_ stays below E_Cl_. Accordingly [Cl^−^]_i_ shows only a small transient change at low g_pas_, while a steady decline in [Cl^−^]_i_ occurs at high g_pas_.

**Figure 2 ijms-20-01416-f002:**
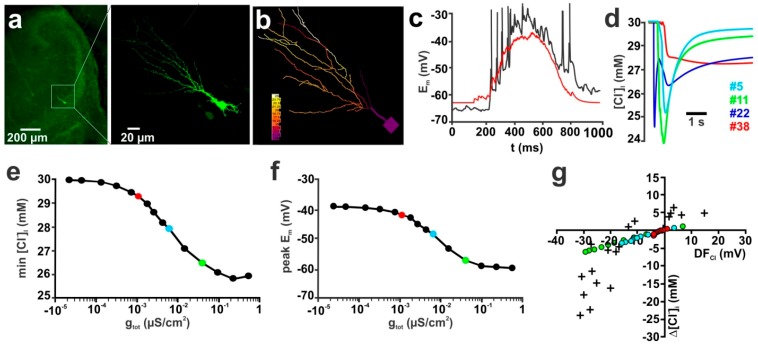
Passive membrane conductance (g_pas_) influences GABA-induced [Cl^−^]_i_ transients in a reconstructed CA3 pyramidal neuron. Similar to the simulations in the isolated dendrite (Figure 1), GABAergic depolarization during a GDP approaches ECl at low g_pas_ values, thereby minimizing the driving force for Cl^−^ fluxes. (**a**) Immunofluorescence image of a biocytin labeled CA3 pyramidal neuron. (**b**) Reconstruction of this CA3 neuron as instrumented for NEURON simulation, with the colors representing the [Cl^−^]_i_ during an exemplary GDP. (**c**) Typical E_m_ trace of a GDP recorded in a real CA3 pyramidal neuron (black trace) and a simulated E_m_ trace of the reconstructed neuron upon stimulation with GDP-derived parameters (red trace). (**d**) Representative [Cl^−^]_i_ transients during a GDP displayed for 4 arbitrary dendrites. Note the asynchronous onset of individual [Cl^−^]_i_ transients and that [Cl^−^]_i_ transients are composed of synaptic Cl^−^ influx and diffusion from adjacent elements. (**e**) The average dendritic [Cl^−^]_i_ depends on the total conductance (g_tot_) of the simulated cell. Please note that a cell that resembles the passive conductance of an immature hippocampal neurons (red symbol: R_Input_ = 901 MΩ) shows only a marginal [Cl^−^]_i_ decrease, while in cells equipped with a mature g_pas_ (cyan symbol: R_Input_ = 189 MΩ, green symbol: R_Input_ = 41 MΩ) larger GPD-induced [Cl^−^]_i_ transients occur. (**f**) Effect of g_tot_ on the peak depolarization during a GDP. Symbols are marked as indicated in (**e**). (**g**) Relationship between GABAergic driving force (DF_Cl_) and GDP-induced [Cl^−^]_i_ transients. The crosses mark values determined experimentally in real CA3 pyramidal neurons. The colored cycles displays the [Cl^−^]_i_ changes computed for the three given R_Input_ values as indicated in (**e**). Note that for the immature R_Input_ only negligible GPD-induced [Cl^−^]_i_ changes are generated (a, b and c modified and used with permission from [45]).

**Figure 3 ijms-20-01416-f003:**
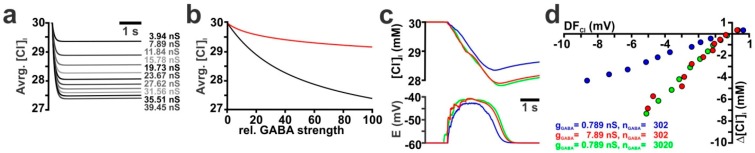
Influence of the GABAergic conductance (g_GABA_) on GABA-induced [Cl^−^]_i_ transients. (**a**) Time course of average [Cl^−^]_i_ in an isolated dendrite upon single synaptic stimulation using g_GABA_ between 3.945 nS and 39.45 nS. (**b**) Average [Cl^−^]_i_ in an isolated dendrite stimulated at a single synapse with g_GABA_ between 0.789 nS (red trace) and 78.9 nS (black trace). Note the non-linear dependency between [Cl^−^]_i_ changes and g_GABA_. In an additional set of simulations, the total GABAergic current was varied by increasing the number (n_GABA_) of evenly distributed single synapses (with g_GABA_ = 0.789 nS) from 1 to 100 (red trace). Note that under these conditions, smaller [Cl^−^]_i_ changes occur. (c) GDP-induced average [Cl^−^]_i_ and E_m_ changes in a reconstructed CA3 pyramidal neuron under control conditions (τ = 37 ms, P_HCO_3__ = 0, g_GABA_ = 0.789 nS, n_GABA_ = 302, blue trace) and upon enhanced stimulation by either increasing the conductance (g_GABA_ = 7.89 nS, n_GABA_ = 302, red trace) or the number of synapses (g_GABA_ = 0.789 nS, n_GABA_ = 3020, green trace). (**d**) Dependency between DF_Cl_ and the GDP-induced [Cl^−^]_i_ transients obtained with different stimulation conditions.

**Figure 4 ijms-20-01416-f004:**
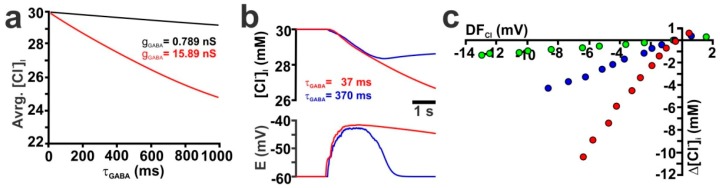
Influence of the decay time constant of GABA receptors (τ_GABA_) on GABA-induced [Cl^−^]_i_ transients. (**a**) Relationship between average [Cl^−^]_i_ and τ_GABA_ at g_GABA_ of 0.789 nS (black trace) or 15.78 nS (red trace) upon a single synaptic stimulation (P_HCO_3__ = 0, [Cl^−^]_i_ = 30 mM) in an isolated dendrite. (**b**) GDP-induced average [Cl^−^]_i_ and E_m_ changes (n_GABA_ = 302, g_GABA_ = 0.789 nS, P_HCO_3__ = 0) using τ_GABA_ of 37 ms (red trace) and 370 ms (blue trace) in a reconstructed CA3 pyramidal neuron. (**c**) Relationship between DF_Cl_ and the GDP-induced [Cl^−^]_i_ transients obtained with different τ_GABA_ of 3.7 ms (green), 37 ms (blue) and 370 ms (red).

**Figure 5 ijms-20-01416-f005:**
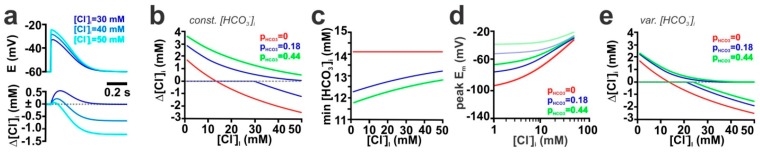
Influence of the relative HCO_3_^−^ conductivity (P_HCO_3__) on GABA-induced membrane depolarization and [Cl^−^]_i_ transients in an isolated dendrite. Activity-dependent decline in [HCO_3_^−^]_i_ reduces GABAergic depolarization and affects [Cl^−^]_i_ changes. (**a**) Time course of E_m_ and [Cl^−^]_i_ changes (Δ[Cl^−^]_i_) upon a single synaptic stimulation (g_GABA_ = 7.89 nS, τ = 37 ms, P_HCO_3__ = 0.18, [HCO_3_^−^]_i_ = 14.1 mM) at initial [Cl^−^]_i_ of 30 mM (dark blue), 40 mM (middle) and 50 mM (light blue). Note that at intermediate [Cl^−^]_i_, a synaptic stimulus can induce biphasic [Cl^−^]_i_ responses. (**b**) Dependency between Δ[Cl^−^]_i_ and [Cl^−^]_i_ upon a single synaptic stimulation (g_GABA_ = 7.89 nS, τ = 37 ms, [HCO_3_^−^]_i_ = 14.1 mM) for different P_HCO_3__. Note the biphasic responses for P_HCO_3__ of 0.18 (represented by the two blue lines) and that at higher P_HCO_3__ the [Cl^−^]_i_ fluxes are shifted towards influx even for high initial [Cl^−^]_i_. (**c**) Dependency between [HCO_3_^−^]_i_ and [Cl^−^]_i_ upon a single synaptic stimulation using a model with dynamic [HCO_3_^−^]_i_ (g_GABA_ = 7.89 nS, τ = 37 ms, initial [Cl^−^]_i_ = 30 mM, initial [HCO_3_^−^]_i_ = 14.1 mM). (**d**) Dependency between peak depolarization and [Cl^−^]_i_ upon a single synaptic stimulation (conditions as in c) at different P_HCO_3__. Note that the implementation of dynamic [HCO_3_^−^]_i_ (plain lines) massively reduces peak depolarization as compared to conditions with static [HCO_3_^−^]_i_ (shaded lines). (**e**) Dependency between [Cl^−^]_i_ changes and [Cl^−^]_i_ upon a single synaptic stimulation (conditions as in **c**). Dual lines with identical colors represent biphasic responses. Note the reduced [Cl^−^]_i_ changes with dynamic [HCO_3_^−^]_i_ as compared to static [HCO_3_^−^]_i_ conditions (shown in b) and that the [Cl^−^]_i_ at which Cl^−^ influx changes to Cl^−^ efflux was shifted to lower [Cl^−^]_i_.

**Figure 6 ijms-20-01416-f006:**
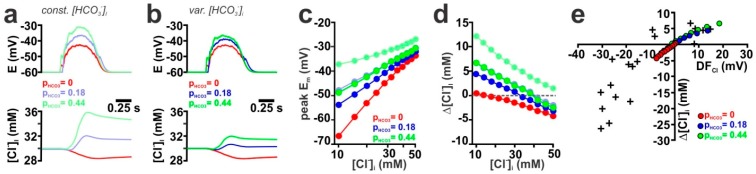
Influence of P_HCO_3__ on GABA-induced [Cl^−^]_i_ transients in a reconstructed CA3 pyramidal neuron. (**a**) Time course of E_m_ and average [Cl^−^]_i_ during a simulated GDP at different P_HCO_3__ (g_GABA_ = 0.789 nS, initial [Cl^−^]_i_ = 30 mM, [HCO_3_^−^]_i_ = 14.1) using a model with a constant [HCO_3_^−^]_i_. (**b**) E_m_, average [Cl^−^]_i_ and [HCO_3_^−^]_i_ during a simulated GDP at different P_HCO_3__ (g_GABA_ = 0.789 nS, initial [Cl^−^]_i_ = 30 mM,) using a model that implements dynamic [HCO_3_^−^]_i_. Note that membrane depolarization and [Cl^−^]_i_ transients are diminished upon implementation of dynamic [HCO_3_^−^]_i_. (**c**) Maximal E_m_ during a GDP at different initial [Cl^−^]_i_ and P_HCO_3__ using static (shaded lines) or dynamic [HCO_3_^−^]_i_ (plain lines). (**d**) GDP-induced [Cl^−^]_i_ changes at different initial [Cl^−^]_i_ and P_HCO_3__ using static (shaded lines) or dynamic [HCO_3_^−^]_i_ (plain lines). (**e**) Dependency between DF_Cl_ and the GDP-induced [Cl^−^]_i_ transients obtained with different P_HCO_3__ under dynamic [HCO_3_^−^]_i_ conditions at P_HCO_3__ of 0 (red), 0.18 ms (blue) and 0.44 (green).

**Figure 7 ijms-20-01416-f007:**
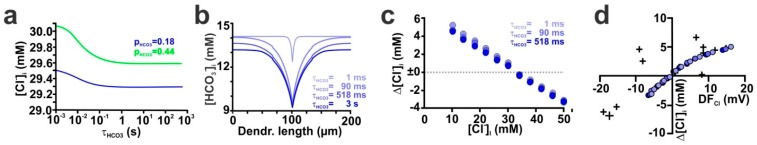
Influence of the stability of HCO_3_^−^ gradients (via variations in τ_HCO_3__) on GABA-induced membrane depolarization and [Cl^−^]_i_ transients. (**a**) Dependency between [Cl^−^]_i_ changes (determined 1 s after stimulus) and τ_GABA_ at P_HCO_3__ of 0.18 and 0.44 upon a single synaptic stimulation (g_GABA_ = 7.89 nS, τ = 37 ms, P_HCO_3__ = 0.18, initial [Cl^−^]_i_ = 30 mM) in an isolated dendrite. Note that at τ_HCO_3__ of ca. 1 s the maximal [Cl^−^]_i_ changes are reached. (**b**) Spatial profile of maximal [HCO_3_^−^]_i_ changes upon the single synaptic stimulation (parameters as in a) at different τ_HCO_3__. Note that τ_HCO_3__ influences the spatial profile of [HCO_3_^−^]_i_, although the peak [HCO_3_^−^]_i_ values are mainly comparable. (**c**) Dependency between maximal GDP-induced [Cl^−^]_i_ changes (g_GABA_ = 0.789 nS, τ = 37 ms, P_HCO_3__ = 0.18, n_GABA_ = 395, initial [Cl^−^]_i_ = 30 mM) and initial [Cl^−^]_i_ for different τ_HCO_3__ in the reconstructed CA3 pyramidal neuron. Note that the influence of τ_HCO_3__ on [HCO_3_^−^]_i_ changes is largest at low [Cl^−^]_i_, but that overall τ_HCO_3__ has only a minimal impact on the [Cl^−^]_i_ changes. (**d**) Dependency between DF_Cl_ and the GDP-induced [Cl^−^]_i_ transients obtained with different τ_HCO_3__ (shadings as in **c**).

**Figure 8 ijms-20-01416-f008:**
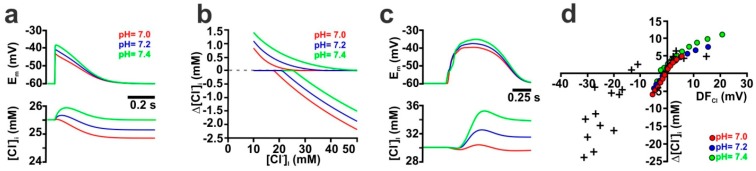
Influence of pH on GABA-induced membrane depolarization and [Cl^−^]_i_ transients. (**a**) Time course of E_m_ and [Cl^−^]_i_ upon a single synaptic stimulation (g_GABA_ = 7.89 nS, τ = 37 ms, P_HCO_3__ = 0.18, initial [Cl^−^]_i_ = 30 mM) in an isolated dendrite at different pH. Note the effect of pH on the depolarizations and that the biphasic [Cl^−^]_i_ at pH 7.2 was transformed into Cl^−^ efflux at pH 7.0 and to Cl^−^ influx at pH 7.4. (**b**) Dependency between GDP-induced [Cl^−^]_i_ changes and initial [Cl^−^]_i_ for different pH. For each pH the two lines represent maximal and minimal [Cl^−^]_i_ changes. Note that pH 7.0 shifts [Cl^−^]_i_ changes towards Cl^−^ efflux, whereas pH 7.4 shifts [Cl^−^]_i_ changes towards Cl^−^ influx. (**c**) Time course of GDP-induced depolarization and [Cl^−^]_i_ changes in the reconstructed CA3 pyramidal neuron (g_GABA_ = 0.789 nS, τ = 37 ms, P_HCO_3__ = 0.18, n_GABA_ = 395, initial [Cl^−^]_i_ = 30 mM) at different pH (color code as in a). Note that the GDP-induced [Cl^−^]_i_ changes are diminished at pH 7.0 and enhanced at pH 7.4. (**d**) Dependency between DF_Cl_ and the GDP-induced [Cl^−^]_i_ transients obtained at different pH. Note that at pH 7.0 the GDP-induced [Cl^−^]_i_ increase was diminished, while the [Cl^−^]_i_ decrease was slightly enhanced.

**Figure 9 ijms-20-01416-f009:**
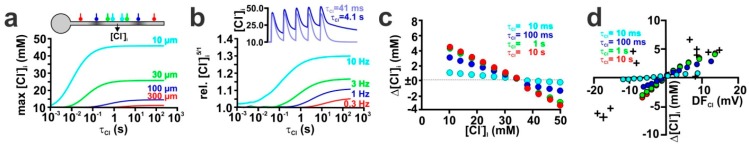
Influence of Cl^−^ diffusion and the kinetics of transmembrane Cl^−^ transport on GABA-induced [Cl^−^]_i_ transients. (**a**) Dependency between τ_Cl_ and [Cl^−^]_i_ determined in the middle between 2 simultaneously stimulated synapses (parameters as in a) located 10 µM, 30 µM, 100 µM and 100 µm from the node of [Cl^−^]_i_ recording. The inset represents a schematic illustration of the spatial arrangement. (**b**) Analysis of temporal summation of activity-dependent [Cl^−^]_i_ transients upon 5 consecutive GABA stimuli (parameters as in **a**) provided at frequencies of 0.3 Hz, 1 Hz, 3 Hz and 10 Hz in the dendrite + soma arrangement. The inset illustrates typical [Cl^−^]_i_ traces obtained at 3 Hz with τ_Cl_ of 41 ms and 4.1 s. The ratio in the [Cl^−^]_i_ between the first and fifth stimulus (rel. [Cl^−^]_i_^5/1^) shows a sigmoidal dependency on τ_Cl_. Note that with higher stimulus frequencies faster τ_Cl_ are required to prevent summation of [Cl^−^]_i_ transients. (**c**) Dependency between maximal GDP-induced [Cl^−^]_i_ changes (g_GABA_ = 0.789 nS, τ_GABA_ = 37 ms, P_HCO_3__ = 0.18, τ_HCO_3__ = 1 s, n_GABA_ = 395) and initial [Cl^−^]_i_ for different τ_Cl_ in the reconstructed CA3 neuron (**d**) Dependency between DF_Cl_ and the GDP-induced [Cl^−^]_i_ transients obtained with different τ_Cl._

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
