# Peer review of "Interactions between Membrane Resistance, GABA-A Receptor Properties, Bicarbonate Dynamics and Cl-Transport Shape Activity-Dependent Changes of Intracellular Cl Concentration"

_ijms, 2019, doi:10.3390/ijms20061416_

Round 1

Reviewer 1 Report

The authors carry out compartmental modeling to study intracellular Cl-
concentration ([Cl-]i) changes during GABAergic synaptic transmission both
in an isolated "ball and stick" dendrite and in a reconstructed immature CA3 pyramidal neuron that is intended to be representative, using the well-
established NEURON simulation software.  The results suggest that high
resting [Cl-]i, high input resistance, slow decay time of GABA-A receptors,
and dynamic HCO3-gradient are specifically adapted in early postnatal
neurons to facilitate limited activity-dependent [Cl-]i decreases (i.e.
depolarization).  This depolarizing activity of GABA-A channels in
immature cells is unlike the hyperpolarizing (inhibitory) activity in
adult cells, and has been considered somewhat an anomaly.  This study
sheds new light on the nature of this anomaly.

This study provides a very thorough examination of the influence of the
listed factors on Cl-tranport through GABA-A receptors.  In particular,
study of the role of bicarbonate, which is also accommodated by the open
GABA-A channel, sheds light on this often-neglected aspect of the situation.

Without actually attempting to reproduce the numerous simulations, I can
only say that the results generally appear to be in accord with expectation
and the comparison of results for the two models studied provides a kind of
control on the more complex computations with the full-cell model.

I have only the following comments regarding needed additions and
corrections to the manuscript:

Besides those already provided, equations should be given for the main
entities displayed in the figures, particularly [Cl-]i, [HCO3]i, Em,
Ihold, Rinput.

l. 44 refers to a "shorter time scale".  Shorter than what?
l. 66 refers to "attenuation of the [Cl-]i gradient" which would
   suggest a gradient of concentration inside the cell.  Is just
   a [Cl-] gradient (internal minus external) intended?
ll. 225,239,246,342,355 etc. "[Cl-]i" should not be split at the end of
   a line.
Fig. 6d appears to have two separate continuations of the curve for
   PHCO3 = 0.18 at high values of the abscissa; Fig. 6i also for PHCO3 =
   0.44.  Please explain or correct if there is an error in the figures.
ll. 533,538,etc.  What is word "instable"?  "Unstable" intended?
l. 549 "nonlinear" should not be hyphenated.
ll. 708-709 duplicate ll. 706-707.
ll. 715-716 some numbers are missing for quantities d and l.

Author Response

We thank the reviewer for his/her valuable comments that helped us to improve our manuscript. Please find all modification that we made in the manuscript redlined in the revised version of the manuscript.

Besides those already provided, equations should be given for the main entities displayed in the figures, particularly [Cl-]i, [HCO3]i, Em, Ihold, Rinput.

In order to follow this important comment of reviewer #1, we now provided the methods and formulas for the determination of all parameters in the materials and methods section (lines 665ff, 695f, 698f, 718, and 719f). Please also note, that we specified the displayed parameter in the ordinates of several plots, as suggested by reviewer #2 (first minor point), which further specifies the displayed entities.

l. 44 refers to a "shorter time scale".  Shorter than what?

We rephrased this sentence, now stating “activation of GABAA receptors can influence [Cl-]i on a time scale of seconds to minutes” (line 45)

l. 66 refers to "attenuation of the [Cl-]i gradient" which would suggest a gradient of concentration inside the cell.  Is just a [Cl-] gradient (internal minus external) intended?

We thank the reviewer to notice this obvious error. We corrected this statement to [Cl-] gradient throughout the manuscript (also for “[HCO3-] gradient”).

ll. 225,239,246,342,355 etc. "[Cl-]i" should not be split at the end of a line.

Unfortunately these errors did not appear in our version of the PDF. Therefore we assume that this mistyping is related to the processing by the editoral office. As we have no control of these errors and there is no way to protect the brackets  “[“ and “]”) from line wrapping in Word, it is not possible for us to prevent these errors.

Fig. 6d appears to have two separate continuations of the curve for PHCO3 = 0.18 at high values of the abscissa; Fig. 6i also for PHCO3 = 0.44. Please explain or correct if there is an error in the figures.

These figures (now Fig. 5b, e) indeed have to separate continuations for one parameter. This was due to the fact that we plotted the minimal and maximal [Cl-]i changes for biphasic responses separately. We now explicitly mention this fact in the legends to these figures (Fig. 5b, e, line 257 and line 264; Fig. 8b, line 375) and also added a sentence to the results part to emphasize that “we determined for such biphasic responses the maximal and minimal [Cl-]i upon GABAergic stimulation [..]” (line 245).

ll. 533,538,etc.  What is word "instable"?  "Unstable" intended?

We corrected this typo (line 549).

l. 549 "nonlinear" should not be hyphenated.

We corrected this error (line 470).

ll. 708-709 duplicate ll. 706-707.

We rephrased these sentences to avoid duplicate information (lines 639-641).

ll. 715-716 some numbers are missing for quantities d and l.

We added these numbers (lines 641-642).

Reviewer 2 Report

@page { margin: 0.79in } p { margin-bottom: 0.1in; line-height: 120% } a:link { so-language: zxx }

In this work the authors perform an extensive parameter search to elucidate the key determinants regulating the equilibrium of the internal chloride concentration that in turn determines the efficacy of inhibitory neurotransmission. Their work builds on previous experimental findings of the same laboratory and it extends nicely previous modeling works. However I found the reading of the manuscript a bit difficult given the numerous results and plots along the manuscript.

I outline my review as follows with some suggestions for improving their work.

Major points:

1) As mentioned before I strongly recommend to the authors to keep a simpler outline of their results in the main body of the manuscript and move the rest of their results to a Supporting Information section.

2) The authors relate the low/high input resistance to the degree of maturation of the neural tissue and this is an interesting point. Strangely enough, the authors recognize, in the Discussion section, the importance of the the effect of H+ transport and the exchangers for chloride homeostasis which importance has already been highlighted in previous modeling works (Doyon et al. Plos Comp Biol 2011). Based on that work one could expect a 10 mV change of the chloride reversal potential , during a repetitive stimulation protocol, when that mechanism is incorporated in the model. Here I would like to see the importance of the additional mechanism on the GDPs.

3) I have not seen any sentence in the manuscript about the sharing policy of your code upon publication of you paper. I suppose you are going to release the code at least to ModelDB (https://senselab.med.yale.edu/modeldb/).

Minor:

Since there are many results in the paper it is somehow difficult to go through all panels with a clear understanding on what quantities are reported. For example, when dealing with maximal chloride (line 469) I suggest to report it explicitly in the label of the axis.

lines 123-124: “resembles the distribution of GABAergic inputs during a GDP” sounds a bit vague to me, please be more precise

line 258: “with 0.789 nS and ...” should be “with 0.789 nS (red) and ...”

line 290: “(blue trace)” and “(red trace)” are reversed

Figure 7c, Figure 7f dotted gray line is barely visible, improve the quality

line 419: “effected” is “affected”

line 461: 1) Figure 9e is Figure 9f 2) to me here it seems that chloride transients are similarly abolished in Fig.9f Fig.9d and Fig.9e

line 555: “provide some protection” what do you mean here?

line 557: there is no panel 1g !

line 630: “an” is “a”

Author Response

We thank the reviewer for his/her valuable comments that helped us to improve our manuscript. Please find all modification that we made in the manuscript redlined in the revised version of the manuscript.

Major points:

1) As mentioned before I strongly recommend to the authors to keep a simpler outline of their results in the main body of the manuscript and move the rest of their results to a Supporting Information section.

In order to follow the reviewer’s suggestion we restructured the results part and shifted less important panels to supplementary figures 1-4. Some of the quantitative results are eliminated from the main manuscript and now provided in the legends to the supplementary figures.

2) The authors relate the low/high input resistance to the degree of maturation of the neural tissue and this is an interesting point. Strangely enough, the authors recognize, in the Discussion section, the importance of the the effect of H+ transport and the exchangers for chloride homeostasis which importance has already been highlighted in previous modeling works (Doyon et al. Plos Comp Biol 2011). Based on that work one could expect a 10 mV change of the chloride reversal potential , during a repetitive stimulation protocol, when that mechanism is incorporated in the model. Here I would like to see the importance of the additional mechanism on the GDPs.

In order to follow the reviewers suggestion, we simulated the effect of acitic and alkaline pH-shifts on the activity- and GDP-dependent [Cl-]i changes. Due to the design ot our model, it was for us not possible to incorporate the complex multiparametric model used by Doyon et al. within the limited time (originally 5 days) given by the editors. However, our results (lines 344-369) demonstrated that an acidic pH-shift indeed dampens the activity-dependent [Cl-]i transients. These results are illustrated in the new Fig. 8 and discussed in lines 527-536.

Regarding the contribution of a HCO3-/Cl- shift via the anion exchanges, we consider that the contribution of this HCO3-/Cl- exchange to activity-dependent [Cl-]i transients is in immature neurons rather limited, as the time course of [Cl-]i regulation is in the range of minutes and thus the elimination of activity-dependent [Cl-]i and [HCO3-]i transients is dominated by diffusion. And as the anion exchanger contribute less that the NKCC1 to [Cl-]i homeostasis, we assume that the impact of this transporter to [Cl-]i transients can be neglected. In addition, also for this parameter the design of our model disables the simple incorporation of such a process to our simulation. We discussed this issue in the revised version of the manuscript (line 588).

3) I have not seen any sentence in the manuscript about the sharing policy of your code upon publication of you paper. I suppose you are going to release the code at least to ModelDB (https://senselab.med.yale.edu/modeldb/).

We uploaded the final models to the Model DB Database and also included them as supplementary material with this manuscript. (see lines 633-635)

Minor:

Since there are many results in the paper it is somehow difficult to go through all panels with a clear understanding on what quantities are reported. For example, when dealing with maximal chloride (line 469) I suggest to report it explicitly in the label of the axis.

We optimized the axis labels as suggested by the reviewer.

lines 123-124: “resembles the distribution of GABAergic inputs during a GDP” sounds a bit vague to me, please be more precise

In order to follow the reviewers concern we now provide more details on the stimulation parameters as follows: “Each of this 101 GABAergic synapses was randomly distributed within the dendrites of the reconstructed neuron. The time point for the stimulation of each synapse was given by a normal distribution (µ = 600ms, σ = 900 ms). These values were derived from in-vitro experiments and resembles the distribution of GABAergic inputs during a GDP (Lombardi et al, 2018)” (lines 124-128). Please note that we also provided more details on this issue in the materials and methods section (lines 661-663).

line 258: “with 0.789 nS and ...” should be “with 0.789 nS (red) and ...”

We included this information (Legend to Fig. 3b, line 207).

line 290: “(blue trace)” and “(red trace)” are reversed

We corrected this error (Legend to Fig. 4b, line 230)

Figure 7c, Figure 7f dotted gray line is barely visible, improve the quality

We enhanced the visibility of this line, (now in figure 6d and figure S4c).

line 419: “effected” is “affected”

We corrected this typo (line 327).

line 461: 1) Figure 9e is Figure 9f 2) to me here it seems that chloride transients are similarly abolished in Fig.9f Fig.9d and Fig.9e

These figures are now presented as supplementary figures S4i-k and the quantification has been removed from the manuscript. Therefore the wrong reference to panel 9e has been removed. The differences in the [Cl-]i distribution between panels 9d-f) (Now supplementary figure S4i-k) is only visible in the traces for 1 and 4 s. To follow the reviewers concern, we now explicitly stated this information in the legend to these figures (lines 1050-1053). 

line 555: “provide some protection” what do you mean here?

We thank the reviewer for pointing us to this error. To follow the reviewers hint, we rephrased this paragraph now stating: “At low R(Input) the passive membrane conductance stabilizes Em and thus DF(Cl). In consequence, larger Cl- fluxes can be expected. Accordingly, implementation of “adult like” membrane properties [47] in a reconstructed immature neuron massively enhanced activity-dependent [Cl-]i changes (Figure 1c). In contrast, it seems obvious that immature neurons, with their high R(Input) [59], are less susceptible to activity-dependent [Cl-]i changes.” (line 492-497).

line 557: there is no panel 1g !

We corrected this typo (line 476)

line 630: “an” is “a”

We corrected this typo.